# Permutation-based identification of important biomarkers for complex diseases via machine learning models

Xinlei Mi [1], Baiming Zou[2], Fei Zou [2] & Jianhua Hu [3✉]

Study of human disease remains challenging due to convoluted disease etiologies and complex molecular mechanisms at genetic, genomic, and proteomic levels. Many machine learning-based methods have been developed and widely used to alleviate some analytic challenges in complex human disease studies. While enjoying the modeling flexibility and robustness, these model frameworks suffer from non-transparency and difficulty in interpreting each individual feature due to their sophisticated algorithms. However, identifying important biomarkers is a critical pursuit towards assisting researchers to establish novel hypotheses regarding prevention, diagnosis and treatment of complex human diseases. Herein, we propose a Permutation-based Feature Importance Test (PermFIT) for estimating and testing the feature importance, and for assisting interpretation of individual feature in complex frameworks, including deep neural networks, random forests, and support vector machines. PermFIT (available at https://github.com/SkadiEye/deepTL) is implemented in a computationally efficient manner, without model refitting. We conduct extensive numerical studies under various scenarios, and show that PermFIT not only yields valid statistical inference, but also improves the prediction accuracy of machine learning models. With the application to the Cancer Genome Atlas kidney tumor data and the HITChip atlas data, PermFIT demonstrates its practical usage in identifying important biomarkers and boosting model prediction performance.

[1] Department of Preventive Medicine,  Northwestern University, Feinberg School of Medicine, Chicago, IL, USA. [2] Department of Biostatistics, University of North Carolina at Chapel Hill, Chapel Hill, NC, USA. [3] Department of Biostatistics and Department of Medicine, Columbia University, New York, NY, USA. ✉email: jh3992@cumc.columbia.edu

With the advancement of high-throughput technologies, massive amounts of high-dimensional omics data have been generated and made available through large public databases due to great data sharing efforts by the research community, such as The Cancer Genome Atlas (TCGA)[1]. These data are valuable in elucidating the molecular mechanisms of disease phenotypes[2,3]. However, study of complex human disease remains challenging due to convoluted disease etiologies and underlying intricate molecular mechanisms at genetic, genomic, and proteomic levels. Many popular machine learning algorithms, such as non-linear kernel support vector machines (SVMs), random forests (RFs), and deep neural networks (DNNs) in artificial intelligence areas, have been developed to build more powerful predictive models for biomedical and bio-omics data regarding clinical outcomes, e.g. drug response[4], and medical imaging classification[5]. While enjoying the modeling flexibility and robustness, these model frameworks suffer from non-transparency and difficulty in interpreting the role of each individual feature due to their sophisticated algorithms, compared with those more interpretable parametric models, such as linear regressions, logistic regressions, and decision trees. Nonetheless, identifying important biomarkers associated with complex human disease is a critical pursuit towards assisting researchers to establish novel hypotheses regarding prevention, diagnosis and treatment of complex human diseases. Accurate identification of important biomarkers associated with complex human disease not only provides valuable insights into their underlying genetic architecture and disease etiology but also offers great potentials for early disease diagnosis, improved precision medicine, innovative treatment development, and accurate disease risk and progression prediction[6].

To address the non-transparency in the association study between disease outcomes and predictors using machine learning models, the feature importance score strategy has been proposed and extensively investigated[7–13], including surrogate models[7,14], Shapley value-based methods[15,16], conditional randomization tests (CRTs)[10], knockoff models (i.e., model-X)[10,12], and permutation-based feature importance[8]. Surrogate modeling methods approximate the complex models by using explanatory surrogate models, such as linear models or decision trees. While enjoying the great flexibility in choosing the surrogate models, the feature importance is still restricted to the selected explanatory models that might be misspecified[13]. Shapley value-based methods, such as SHAP[16], provide localized feature characterization based on game theory, while they are computationally intensive and do not guarantee a valid test. Both CRT and model-X knockoff were proposed in Candes et al.[10], while CRT is less preferred due to its expensive computational cost. Model-X knockoff is more computationally efficient in performing feature importance test via constructing knockoff features. Recently, model-X knockoff was adopted for DNN[12] models. Tansey et al.[11] proposed the holdout randomization test (HRT) to reduce the computational cost of CRT via avoiding model refitting.

The overall disadvantage of CRT, HRT, and model-X knockoff is that they all depend on the assumption of a known covariance structure[10]. When the covariance structure is not accurately estimated, their performance could be severely impacted[17]. Although KnockoffGAN[18], an extension of model-X knockoff, does not suffer such disadvantage, it is difficult to train adversarial networks[19] and requires more tuning. Another strategy to avoid suffering from the known covariance structure assumption is approaches based on Gaussian mirrors[20–22]. Specifically, Xing et al.[21] proposed individual neural Gaussian mirror (INGM) and simultaneous neural Gaussian mirror (SNGM). However, INGM requires repetitive model fitting, which is computationally costly, while SNGM is efficient but could suffer performance loss[21]. The

permutation-based feature importance learning method, another popular approach for feature selection, measures the change of prediction errors due to the shuffling of a feature. The larger the increase of prediction errors is, the more impact a feature makes on the outcome of interest. However, unlike CRT, HRT, or model-X knockoff, permutation-based feature selection does not require prior knowledge of feature distributions and thus it is more statistically robust. Several permutation-based feature importance methods have been proposed, with applications mainly on random forests and DNNs[8,9,23]. These methods either do not conduct any statistical inference or cannot offer valid inference on the feature importance. For example, Putin et al.[23] applied permutation-based importance scores to DNNs to identify biomarkers associated with human aging, but provided no formal statistical testing. Notably, Altmann et al.[9] proposed a corrected permutation-based importance score approach for random forest, which however, is difficult to be generalized to other machine learning model frameworks.

To overcome the aforementioned challenges, we propose a general permutation-based feature importance test (abbreviated as PermFIT), for complex machine learning models, which takes advantage of (i) permutation test coupled with cross-fitting to obtain a valid importance score test that properly controls the type-I error; and (ii) selecting important features from PermFIT to further improve the accuracy of these predictive models. We implement PermFIT for the following machine learning models, including DNN, RF, and SVM. More specifically, PermFIT first approximates the function that maps features to the outcome, based on which, PermFIT then evaluates the importance score of each feature, defined as the expected increase of prediction errors due to the permutation of the feature. Computationally, the PermFIT framework does not require refitting the models. In order to reduce the bias of important score estimation from the potential model overfitting, we adopt cross-fitting to ensure the validity of the test statistics. PermFIT is motivated by two benchmark data: the Reverse Phase Protein Arrays (RPPAs) data from three kidney cancer studies in The Cancer Genomic Atlas (TCGA) and the HITChip Atlas microbiome data regarding body mass index (BMI). However, PermFIT has broad applicability to a wide variety of biomedical data and more.

## Results

To evaluate the performance of PermFIT, we first conduct comprehensive simulation studies under various scenarios with different sample sizes and correlation structures among features. Moreover, it is applied to two real-world datasets: the Reverse Phase Protein Arrays (RPPA) data from three kidney cancer studies in TCGA and the HITChip Atlas microbiome data. We apply PermFIT to three commonly used machine learning methods: DNN[24,25], RF[8], and SVM[26], denoted as PermFIT-DNN, PermFIT-RF and PermFIT-SVM, respectively. We also compare PermFIT with several existing popular feature selection methods for DNN, RF, and SVM: SHAP[16], LIME[14], holdout randomization test[11], and simultaneous neural Gaussian mirror[21] with DNN (denoted as SHAP-DNN, LIME-DNN, HRT-DNN, and SNGM-DNN, respectively), RF importance evaluation of Breiman[8] (denoted as Vanilla-RF, i.e., an ensemble approach based on decision trees), and SVM with recursive feature elimination[27] (denoted as RFE-SVM). SHAP-DNN, LIME-DNN, and RFE-SVM utilize an importance score to rank input features, from which top features are selected. For each feature, Vanilla-RF provides an importance score estimate and its associated standard error, with which the statistical significance of the feature importance can be tested. HRT provides a p-value for each feature without importance scores. We evaluate these methods as

follows: (i) we apply each method to the training data with all the input features, estimate the feature importance scores, $p$ values, and assess the type-I error; (ii) we refit each model with its corresponding top ranked important variables, and re-evaluate its goodness-of-fit and prediction improvement.

**Simulation studies.** We examine the performance of the proposed methods with the following simulation scenarios. First, we generate the continuous data from the following model,

$$Y = X_1 + 2\log\left(1 + 2X_{p_0+1}^2 + \left(X_{2p_0+1} + 1\right)^2\right) + X_{3p_0+1}X_{4p_0+1} + \epsilon, \quad (1)$$

where $X$ is a $p$-dimensional random variable drawn from MVN(0, $\Sigma$), $p = 10p_0$, $p_0 = 10$, $\Sigma = \text{diag}\{\Sigma_1, \ldots, \Sigma_{10}\}$, is a block-diagonal matrix, $\Sigma_1 = \ldots = \Sigma_{10} = \{\sigma_{ij}\}_{0 < i,j \le p_0}$, are $p_0 \times p_0$ matrices, $\sigma_{ij} = 1$ for $i = j$ and $\sigma_{ij} = \rho$ for $i \ne j$, and $\epsilon \sim N(0, 1)$. $N$ independent observations are drawn from the distribution of $(Y, X)$ in the training set and 10,000 in the test set, which is used to evaluate model fitting performance. To mimic the real-world data, we introduce correlations among variables by blocks, and let one variable from each of the first 5 blocks have a signal. We define $S_0$ and $S_1$ as the sets that contain all the null features that are correlated and uncorrelated with the causal features, respectively, i.e., $S_0 = \{X_j : j \le 5p_0 \text{ and } j \ne 1, p_0 + 1, 2p_0 + 1, 3p_0 + 1, 4p_0 + 1\}$, $S_1 = \{X_j : j > 5p_0\}$. We consider various simulation settings with different values of $\rho \in \{0, 0.2, 0.5, 0.8\}$, and $N \in \{1000, 5000\}$. Each simulation scenario is replicated 100 times.

The results are displayed in Fig. 1 and Table 1. Figure 1a displays detailed feature importance scores generated from each method that we consider. Since HRT does not provide importance scores, we use $-\log_{10}(p \text{ value})$ instead. Note that the estimated importance scores from PermFIT methods and Vanilla RF are in the same scale, while the ones from SHAP-DNN, LIME-DNN and RFE-SVM are not. For $X_1$ whose effect is linear, the importance scores from PermFIT-DNN and PermFIT-SVM are higher, compared with those from RF-based framework due to the restricted tree-based modeling nature of RF. In addition, the RF-based framework can barely detect the interaction between $X_{3p_0+1}$ and $X_{4p_0+1}$ because the split rule in tree-based methods is less effective in dealing with such interactions. Expectedly, as the within-block correlation $\rho$ increases, the estimated importance scores from all methods deviate further away from their estimands. However, PermFIT-SVM remains high power in detecting the true positive features. As $\rho$ increases, it's noticeable that Vanilla-RF and PermFIT-SVM tend to identify the null features that are correlated with the causal features. Compared with Vanilla-RF, PermFIT-RF has fewer false positive discoveries. Overall, PermFIT-DNN provides the most precise and stable importance measure in differentiating the true positive from null features.

The frequency (percentage) of the important variables detected by each method is presented in Table 1. For Vanilla-RF and PermFIT methods that provide p values, the significance level is controlled at 0.05, while for RFE-SVM, the top 10 features with the largest importance scores are selected. First of all, at $\rho = 0$, PermFIT controls the rate of significance findings across all null features at around 0.05, suggesting that the type-I error is well controlled by PermFIT, while Vanilla-RF has the type-I error of 0.09, nearly double of PermFIT. When $N = 1000$, the type-I error of HRT-DNN is slightly inflated. Besides, LIME-DNN and SNGM-DNN show a limited ability in identifying features with nonlinear effects, such as $X_{p_0+1}$, $X_{3p_0+1}$, and $X_{4p_0+1}$. On the other hand, SHAP-DNN is able to assign high rankings to the important features based on the importance scores. However, it fails to offer a valid test for its importance scores; specifically, its

type-I error and power depend on correctly specifying the number of important features. When $\rho$ increases to 0.5 or 0.8, RFE-SVM tends to select the null features that are correlated with the true causal features, or those in $S_0$ more frequently than $X_{3p_0+1}$ and $X_{4p_0+1}$, the two causal variables that interact with each other, demonstrating its limited capability in detecting variables with interaction effects when correlation exists. In contrast, PermFIT-SVM is capable of identifying $X_{3p0+1}$ and $X_{4p0+1}$ consistently at a much higher frequency than all the null features. Compared with PermFIT-RF, Vanilla-RF has a higher power in detecting $X_{3p0+1}$ and $X_{4p0+1}$, but also produces remarkably more false positive findings among features in $S_0$. For example, as $\rho$ goes to 0.8 and $N = 1000$, it results in >80% false positive rate in $S_0$, suggesting a far inferior feature selection performance. In all these scenarios, PermFIT-DNN can consistently identify causal features while controlling false positive findings at a much lower rate than those of Vanilla-RF, PermFIT-RF, and PermFIT-SVM.

Posterior to important feature selection, the prediction performance of the comparing models almost all gets improved. Figure 1b displays the mean squared prediction error (MSPE) of each model, (i) with full input features, respectively denoted as DNN, RF, and SVM; and (ii) with top selected features from PermFIT methods and HRT-DNN at the significance level of 0.1, and top 20 features from SHAP-DNN, LIME-DNN, SNGM-DNN and RFE-SVM. Selected features help boosting the prediction accuracy of all models, except RFE-SVM, LIME-DNN, and SNGM-DNN, across all simulation scenarios. However, LIME-DNN and SNGM-DNN fail to identify certain important features, which leads to deterioration of the model performance. In addition, at $\rho = 0.8$ corresponding to high correlation among within-block input features, RFE-SVM fails to improve the model fitting over SVM because of feature selection failure, in particular, on $X_{3p0+1}$ and $X_{4p0+1}$; its inferior performance to PermFIT-SVM is clearly observed. Moreover, PermFIT-RF outperforms Vanilla-RF in terms of MSPE, because the latter yields more false positives and cannot effectively reduce the feature dimension. We note that PermFIT-DNN and HRT-DNN consistently outperforms all other methods in comparison, due to its high success rate in identifying true positive features while maintaining a considerably low false positive rate at the same time. In particular, PermFIT-DNN has a lower MSPE than that of HRT-DNN when $N = 1000$ and $\rho \le 0.2$, while similar MSPE values in other scenarios.

To further investigate the small sample performance of these methods, we conduct additional simulations with $(N = 300, p = 100)$ and $(N = 500, p = 200)$, and report the results in Table 2. The type-I errors of PermFIT-based methods are not affected much by the change of $N$ and $p$ in these more challenging cases, while those for HRT-DNN are severely inflated, which is likely because, for more challenging data with a smaller $N$ or a larger $p$, HRT-DNN fails to make an accurate estimation of the covariance matrix of the input features.

We further conduct a simulation study on binary outcomes generated from the following model:

$$P(Y = 1|X) = \text{expit}\left(4X_1 + 8\log\left(1 + 2X_{p_0+1}^2 + \left(X_{2p_0+1} + 1\right)^2\right) + 4X_{3p_0+1}X_{4p_0+1} - 11\right),$$
$$(2)$$

where $\text{expit}(x) = 1/(1 + \exp(-x))$. All the other data structures, including $X$, are generated in the same way as in the continuous case. Similar conclusions are observed with the details presented in Supplementary Table 1 and Supplementary Fig. 1.

**TCGA kidney tumor data.** A large collection of clinical and multiple omics data have been made publicly available by TCGA research project[1]. In our analysis, we included three studies of kidney-related cancer types from TCGA research network: kidney

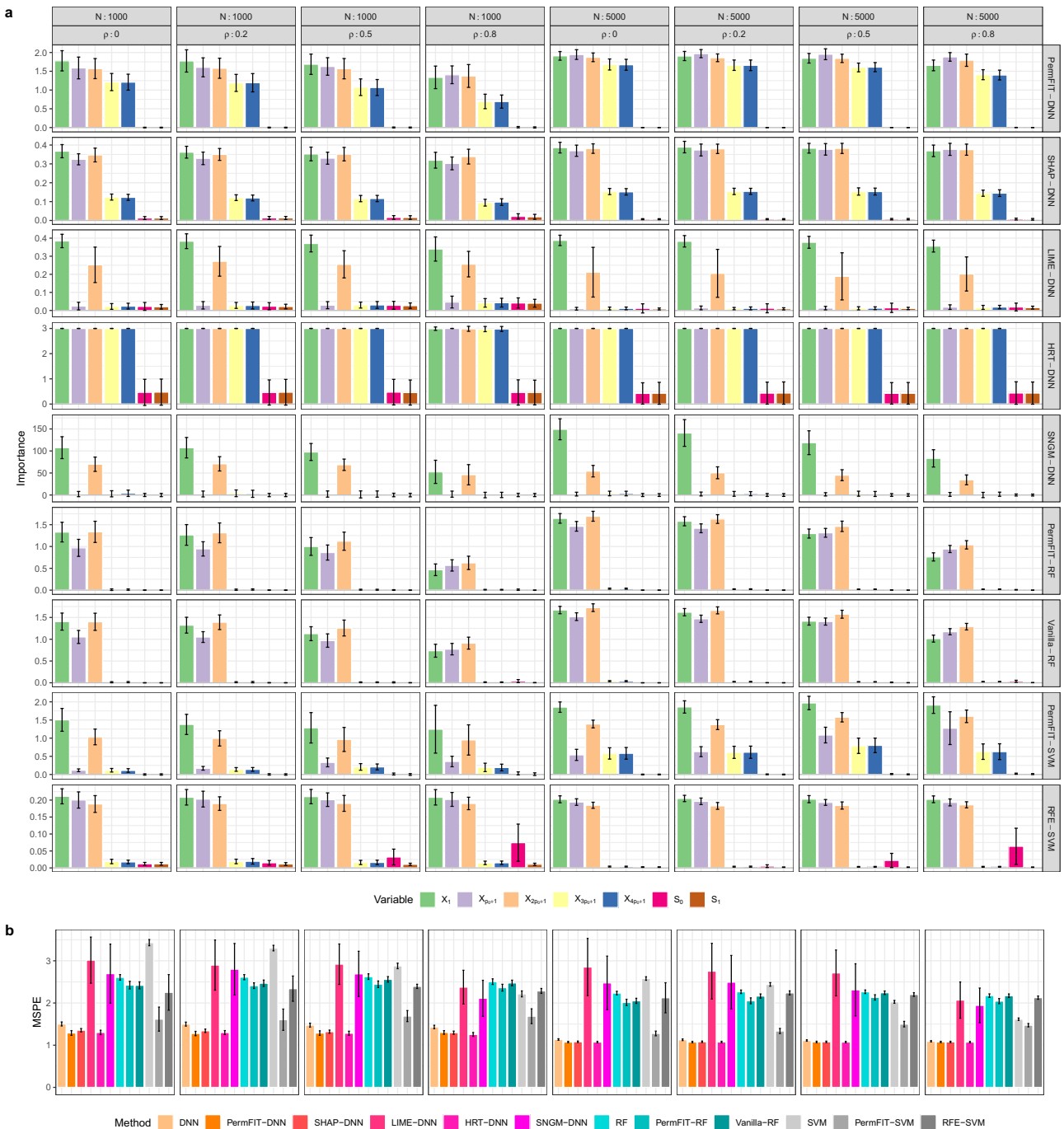

**Fig. 1 Simulation results on continuous outcomes. a** Estimated feature importance for the five true causal features: $X_1$, $X_{p_0+1}$, $X_{2p_0+1}$, $X_{3p_0+1}$, $X_{4p_0+1}$, and two null feature sets: $S_0$ and $S_1$. **b** Mean squared prediction error (MSPE) for methods in comparison. DNN, RF or SVM: specific modeling with all features; PermFIT-DNN, SHAP-DNN, LIME-DNN, HRT-DNN, SNGM-DNN, PermFIT-RF, Vanilla-RF, PermFIT-SVM, or RFE-SVM: specific modeling after feature selection. Data are presented as mean values ± s.d. Simulations in each scenario are repeated for 100 times. Source data are provided as a Source Data file.

renal clear cell carcinoma (KIRC, $N_1 = 537$), kidney renal papillary cell carcinoma (KIRP, $N_2 = 291$), and kidney chromophobe (KICH, $N_3 = 113$). We defined long-term survivor (LTS) as patients who survived more than five years after diagnosis, and short-term survivor (STS) as patients who died within 5 years. We aimed to predict the probability of a patient being in the LTS group and to identify significant biomarkers that contribute to classification of the LTS/STS status. We included 188 LTS and 178 STS subjects with the known survival status in our analysis. We focused our analysis on expression data of 118 proteins

extracted from reverse phase protein arrays (RPPAs)—a highly sensitive, reproducible, and high-throughput proteomic method for protein expression profiling[28].

The negative $\log_{10}(p\ value)$s and the estimated importance scores from each method are presented in Fig. 2 and Supplementary Fig. 3. HRT-DNN, Vanilla-RF and PermFIT models control the FDR at 0.1, and SHAP-DNN, LIME-DNN, SNGM-DNN and RFE-SVM selects 10 features with the largest importance scores. We notice that moderate correlations generally exist among the proteins (see Supplementary Fig. 2).

**Table 1 Simulation results on continuous outcomes.**

| $\rho$ | Variable | N = 1000, p = 100 | | | | | | | | | N = 5000, p = 100 | | | | | | | | |
|---|---|---|---|---|---|---|---|---|---|---|---|---|---|---|---|---|---|---|---|
| | | PermFIT DNN | HRT DNN | PermFIT RF | Vanilla RF | PermFIT SVM | SHAP[a] DNN | LIME[a] DNN | SNGM[a] DNN | RFE[a] SVM | PermFIT DNN | HRT DNN | PermFIT RF | Vanilla RF | PermFIT SVM | SHAP[a] DNN | LIME[a] DNN | SNGM[a] DNN | RFE[a] SVM |
| 0 | $X_1$ | 100 | 100 | 100 | 100 | 100 | 100 | 100 | 100 | 100 | 100 | 100 | 100 | 100 | 100 | 100 | 100 | 100 | 100 |
| | $X_{p_0+1}$ | 100 | 100 | 100 | 100 | 94 | 100 | 17 | 28 | 100 | 100 | 100 | 100 | 100 | 100 | 100 | 13 | 25 | 100 |
| | $X_{2p_0+1}$ | 100 | 100 | 100 | 100 | 99 | 100 | 100 | 100 | 100 | 100 | 100 | 100 | 100 | 100 | 100 | 79 | 100 | 100 |
| | $X_{3p_0+1}$ | 100 | 100 | 26 | 64 | 96 | 100 | 9 | 38 | 41 | 100 | 100 | 98 | 100 | 100 | 100 | 19 | 35 | 42 |
| | $X_{4p_0+1}$ | 100 | 100 | 39 | 71 | 91 | 100 | 12 | 40 | 41 | 100 | 100 | 98 | 100 | 100 | 100 | 17 | 31 | 36 |
| | $S_0$ | 5.4 | 7.5 | 5.3 | 9.4 | 6.1 | 5.1 | 10.2 | 7.3 | 6.5 | 5.0 | 4.9 | 4.9 | 9.3 | 5.0 | 5.7 | 11.5 | 8.1 | 7.2 |
| | $S_1$ | 5.5 | 8.0 | 5.1 | 8.9 | 5.7 | 5.4 | 6.1 | 7.3 | 6.5 | 4.6 | 4.9 | 5.0 | 9.0 | 5.0 | 4.9 | 5.1 | 6.9 | 6.0 |
| 0.2 | $X_1$ | 100 | 100 | 100 | 100 | 100 | 100 | 100 | 100 | 100 | 100 | 100 | 100 | 100 | 100 | 100 | 100 | 100 | 100 |
| | $X_{p_0+1}$ | 100 | 100 | 100 | 100 | 100 | 100 | 19 | 30 | 100 | 100 | 100 | 100 | 100 | 100 | 100 | 25 | 22 | 100 |
| | $X_{2p_0+1}$ | 100 | 100 | 100 | 100 | 100 | 100 | 100 | 100 | 100 | 100 | 100 | 100 | 100 | 100 | 100 | 85 | 100 | 100 |
| | $X_{3p_0+1}$ | 100 | 100 | 36 | 68 | 98 | 100 | 13 | 35 | 27 | 100 | 100 | 94 | 100 | 100 | 100 | 9 | 28 | 0 |
| | $X_{4p_0+1}$ | 100 | 100 | 35 | 69 | 98 | 100 | 15 | 34 | 22 | 100 | 100 | 93 | 100 | 100 | 100 | 11 | 35 | 0 |
| | $S_0$ | 6.1 | 7.1 | 6.7 | 17.5 | 7.9 | 5.1 | 9.9 | 7.6 | 13.2 | 5.7 | 5.2 | 7.5 | 29.0 | 12.9 | 5.3 | 10.9 | 7.8 | 15.6 |
| | $S_1$ | 5.5 | 8.0 | 5.1 | 10.6 | 5.2 | 5.4 | 6.2 | 7.2 | 1.2 | 4.9 | 5.2 | 4.8 | 11.2 | 4.7 | 5.3 | 5.6 | 7.3 | 0.0 |
| 0.5 | $X_1$ | 100 | 100 | 100 | 100 | 100 | 100 | 100 | 100 | 100 | 100 | 100 | 100 | 100 | 100 | 100 | 100 | 100 | 100 |
| | $X_{p_0+1}$ | 100 | 100 | 100 | 100 | 100 | 100 | 13 | 26 | 100 | 100 | 100 | 100 | 100 | 100 | 100 | 17 | 26 | 100 |
| | $X_{2p_0+1}$ | 100 | 100 | 100 | 100 | 100 | 100 | 100 | 100 | 100 | 100 | 100 | 100 | 100 | 100 | 100 | 83 | 100 | 100 |
| | $X_{3p_0+1}$ | 100 | 100 | 35 | 69 | 96 | 100 | 12 | 27 | 0 | 100 | 100 | 96 | 100 | 100 | 100 | 10 | 42 | 0 |
| | $X_{4p_0+1}$ | 100 | 100 | 28 | 68 | 99 | 100 | 11 | 32 | 0 | 100 | 100 | 98 | 100 | 100 | 100 | 10 | 31 | 0 |
| | $S_0$ | 9.3 | 7.5 | 14.1 | 59.3 | 15.9 | 5.5 | 10.8 | 8.0 | 15.6 | 9.6 | 4.9 | 33.8 | 87.8 | 35.2 | 5.6 | 11.5 | 7.5 | 15.6 |
| | $S_1$ | 6.7 | 7.2 | 4.6 | 16.9 | 6.0 | 5.1 | 5.6 | 7.1 | 0.0 | 7.0 | 5.1 | 4.2 | 39.1 | 11.9 | 5.0 | 5.2 | 7.2 | 0.0 |
| 0.8 | $X_1$ | 100 | 100 | 100 | 100 | 100 | 100 | 100 | 100 | 100 | 100 | 100 | 100 | 100 | 100 | 100 | 100 | 100 | 100 |
| | $X_{p_0+1}$ | 100 | 100 | 100 | 100 | 100 | 100 | 16 | 33 | 100 | 100 | 100 | 100 | 100 | 100 | 100 | 22 | 28 | 100 |
| | $X_{2p_0+1}$ | 100 | 100 | 100 | 100 | 100 | 100 | 100 | 100 | 100 | 100 | 100 | 100 | 100 | 100 | 100 | 98 | 100 | 100 |
| | $X_{3p_0+1}$ | 100 | 100 | 54 | 90 | 96 | 99 | 6 | 24 | 0 | 100 | 100 | 99 | 100 | 100 | 100 | 10 | 30 | 0 |
| | $X_{4p_0+1}$ | 100 | 100 | 49 | 88 | 98 | 100 | 10 | 18 | 0 | 100 | 100 | 99 | 100 | 100 | 100 | 14 | 39 | 0 |
| | $S_0$ | 18.8 | 7.2 | 31.6 | 88.3 | 27.7 | 6.2 | 10.7 | 9.0 | 15.6 | 20.7 | 5.7 | 75.3 | 100.0 | 68.2 | 6.6 | 11.9 | 8.4 | 15.6 |
| | $S_1$ | 9.7 | 7.0 | 4.0 | 37.3 | 10.5 | 4.5 | 5.7 | 6.4 | 0.0 | 9.2 | 5.2 | 3.3 | 93.8 | 34.8 | 4.0 | 4.4 | 6.5 | 0.0 |

Reported is the percentage of the important variables detected by each method (p-value cutoff of 0.05), out of 100 repetitions for each simulation scenario, for five true causal features: $X_1$, $X_{p_0+1}$, $X_{2p_0+1}$, $X_{3p_0+1}$, $X_{4p_0+1}$, and two null feature sets: $S_0$ and $S_1$.
aNote that SHAP-DNN, LIME-DNN, SNGM-DNN, and RFE-SVM do not perform formal statistical testing, and features can only be ranked with no associated p values. The reported results for each of these four methods are based on the top 10 selected features for a simple illustration. For PermFIT methods and Vanilla-RF, p values are calculated from one-sided Z test.

**Table 2 Simulation results on continuous outcomes with smaller sample size and/or larger dimension.**

| $\rho$ | Variable | N = 300, p = 100 | | | | | | | | | N = 500, p = 200 | | | | | | | | |
|---|---|---|---|---|---|---|---|---|---|---|---|---|---|---|---|---|---|---|---|
| | | PermFIT DNN | HRT DNN | PermFIT RF | Vanilla RF | PermFIT SVM | SHAP[a] DNN | LIME[a] DNN | SNGM[a] DNN | RFE[a] SVM | PermFIT DNN | HRT DNN | PermFIT RF | Vanilla RF | PermFIT SVM | SHAP[a] DNN | LIME[a] DNN | SNGM[a] DNN | RFE[a] SVM |
| 0 | $X_1$ | 98 | 99 | 99 | 100 | 98 | 100 | 100 | 100 | 100 | 100 | 100 | 100 | 100 | 100 | 100 | 100 | 100 | 100 |
| | $X_{p_0+1}$ | 86 | 91 | 97 | 100 | 22 | 86 | 26 | 27 | 100 | 100 | 100 | 100 | 100 | 15 | 100 | 20 | 34 | 100 |
| | $X_{2p_0+1}$ | 96 | 98 | 100 | 100 | 92 | 100 | 99 | 100 | 100 | 100 | 100 | 100 | 100 | 99 | 100 | 95 | 100 | 100 |
| | $X_{3p_0+1}$ | 54 | 71 | 9 | 19 | 14 | 19 | 14 | 18 | 30 | 83 | 87 | 15 | 27 | 10 | 42 | 15 | 20 | 26 |
| | $X_{4p_0+1}$ | 46 | 61 | 9 | 26 | 18 | 34 | 16 | 18 | 36 | 89 | 96 | 14 | 13 | 10 | 29 | 8 | 13 | 27 |
| | $S_0$ | 5.0 | 16.5 | 5.3 | 9.1 | 4.7 | 7.4 | 9.4 | 8.1 | 6.9 | 5.5 | 22.6 | 5.3 | 7.4 | 5.5 | 3.0 | 5.4 | 3.5 | 3.1 |
| | $S_1$ | 5.5 | 16.7 | 5.4 | 9.2 | 5.8 | 6.5 | 6.4 | 7.5 | 6.4 | 5.6 | 21.5 | 5.1 | 7.0 | 5.5 | 3.4 | 2.5 | 4.0 | 3.5 |
| 0.2 | $X_1$ | 97 | 100 | 98 | 100 | 95 | 100 | 100 | 100 | 100 | 100 | 100 | 100 | 100 | 100 | 100 | 100 | 100 | 100 |
| | $X_{p_0+1}$ | 89 | 93 | 100 | 100 | 37 | 83 | 20 | 22 | 100 | 100 | 100 | 100 | 100 | 29 | 100 | 13 | 31 | 100 |
| | $X_{2p_0+1}$ | 94 | 97 | 98 | 100 | 84 | 100 | 99 | 100 | 100 | 100 | 100 | 100 | 100 | 98 | 100 | 94 | 100 | 100 |
| | $X_{3p_0+1}$ | 56 | 70 | 10 | 24 | 24 | 27 | 21 | 22 | 32 | 93 | 96 | 5 | 19 | 13 | 44 | 10 | 16 | 17 |
| | $X_{4p_0+1}$ | 53 | 69 | 11 | 25 | 21 | 31 | 16 | 18 | 26 | 98 | 95 | 13 | 25 | 15 | 32 | 3 | 13 | 18 |
| | $S_0$ | 5.7 | 16.7 | 5.6 | 11.8 | 6.3 | 7.2 | 9.8 | 8.2 | 8.7 | 6.1 | 22.1 | 5.8 | 11.5 | 6.3 | 3.6 | 5.8 | 3.9 | 5.5 |
| | $S_1$ | 5.9 | 17.6 | 5.4 | 9.5 | 6.4 | 6.7 | 6.0 | 7.4 | 5.0 | 5.9 | 21.9 | 5.0 | 8.1 | 5.6 | 2.8 | 2.3 | 3.7 | 1.4 |
| 0.5 | $X_1$ | 97 | 96 | 94 | 100 | 90 | 100 | 100 | 100 | 100 | 100 | 100 | 99 | 100 | 99 | 100 | 100 | 100 | 100 |
| | $X_{p_0+1}$ | 96 | 98 | 98 | 100 | 66 | 94 | 15 | 24 | 100 | 100 | 100 | 100 | 100 | 52 | 100 | 11 | 33 | 100 |
| | $X_{2p_0+1}$ | 97 | 95 | 99 | 100 | 84 | 100 | 99 | 98 | 100 | 99 | 100 | 100 | 100 | 85 | 100 | 98 | 100 | 100 |
| | $X_{3p_0+1}$ | 67 | 64 | 13 | 18 | 40 | 34 | 17 | 17 | 9 | 92 | 91 | 14 | 23 | 32 | 56 | 7 | 22 | 1 |
| | $X_{4p_0+1}$ | 65 | 66 | 9 | 31 | 37 | 24 | 11 | 14 | 6 | 93 | 97 | 15 | 24 | 33 | 58 | 5 | 16 | 0 |
| | $S_0$ | 9.1 | 17.4 | 8.7 | 31.8 | 9.4 | 7.1 | 9.3 | 8.5 | 14.9 | 8.3 | 21.1 | 9.4 | 32.1 | 10.5 | 3.0 | 5.7 | 4.2 | 7.4 |
| | $S_1$ | 6.6 | 18.1 | 4.7 | 11.5 | 6.3 | 6.5 | 6.8 | 7.2 | 0.3 | 6.0 | 22.3 | 4.7 | 10.4 | 6.2 | 3.0 | 2.4 | 3.3 | 0.0 |
| 0.8 | $X_1$ | 90 | 87 | 69 | 100 | 83 | 100 | 94 | 96 | 97 | 100 | 92 | 82 | 100 | 95 | 100 | 93 | 100 | 100 |
| | $X_{p_0+1}$ | 85 | 89 | 95 | 100 | 65 | 95 | 11 | 20 | 91 | 100 | 98 | 99 | 100 | 64 | 100 | 8 | 31 | 94 |
| | $X_{2p_0+1}$ | 90 | 89 | 73 | 100 | 75 | 100 | 80 | 94 | 90 | 99 | 98 | 90 | 100 | 89 | 100 | 84 | 97 | 83 |
| | $X_{3p_0+1}$ | 62 | 58 | 16 | 33 | 44 | 34 | 11 | 20 | 0 | 80 | 85 | 16 | 40 | 48 | 42 | 6 | 21 | 0 |
| | $X_{4p_0+1}$ | 63 | 60 | 14 | 39 | 48 | 17 | 7 | 18 | 0 | 81 | 79 | 21 | 42 | 46 | 44 | 2 | 20 | 0 |
| | $S_0$ | 16.4 | 17.7 | 17.8 | 67.7 | 19.1 | 8.5 | 10.3 | 10.2 | 16.0 | 13.4 | 21.4 | 18.9 | 67.7 | 18.8 | 4.2 | 5.6 | 5.0 | 7.6 |
| | $S_1$ | 7.4 | 17.5 | 4.6 | 18.2 | 5.9 | 5.4 | 6.6 | 5.6 | 0.0 | 7.1 | 22.1 | 4.4 | 17.2 | 9.5 | 2.2 | 2.8 | 2.5 | 0.0 |

Reported is the percentage of the important variables detected by each method ($p$ value cutoff of 0.05), out of 100 repetitions for each simulation scenario, for five true causal features: $X_1$, $X_{p_0+1}$, $X_{2p_0+1}$, $X_{3p_0+1}$, $X_{4p_0+1}$, and two null feature sets: $S_0$ and $S_1$.
[a]Note that SHAP-DNN, LIME-DNN, SNGM-DNN, and RFE-SVM do not perform formal statistical testing, and features can only be ranked with no associated $p$ values. The reported results for each of these four methods are based on the top 10 selected features for a simple illustration. For PermFIT methods and Vanilla-RF, $p$ values are calculated from one-sided Z test.

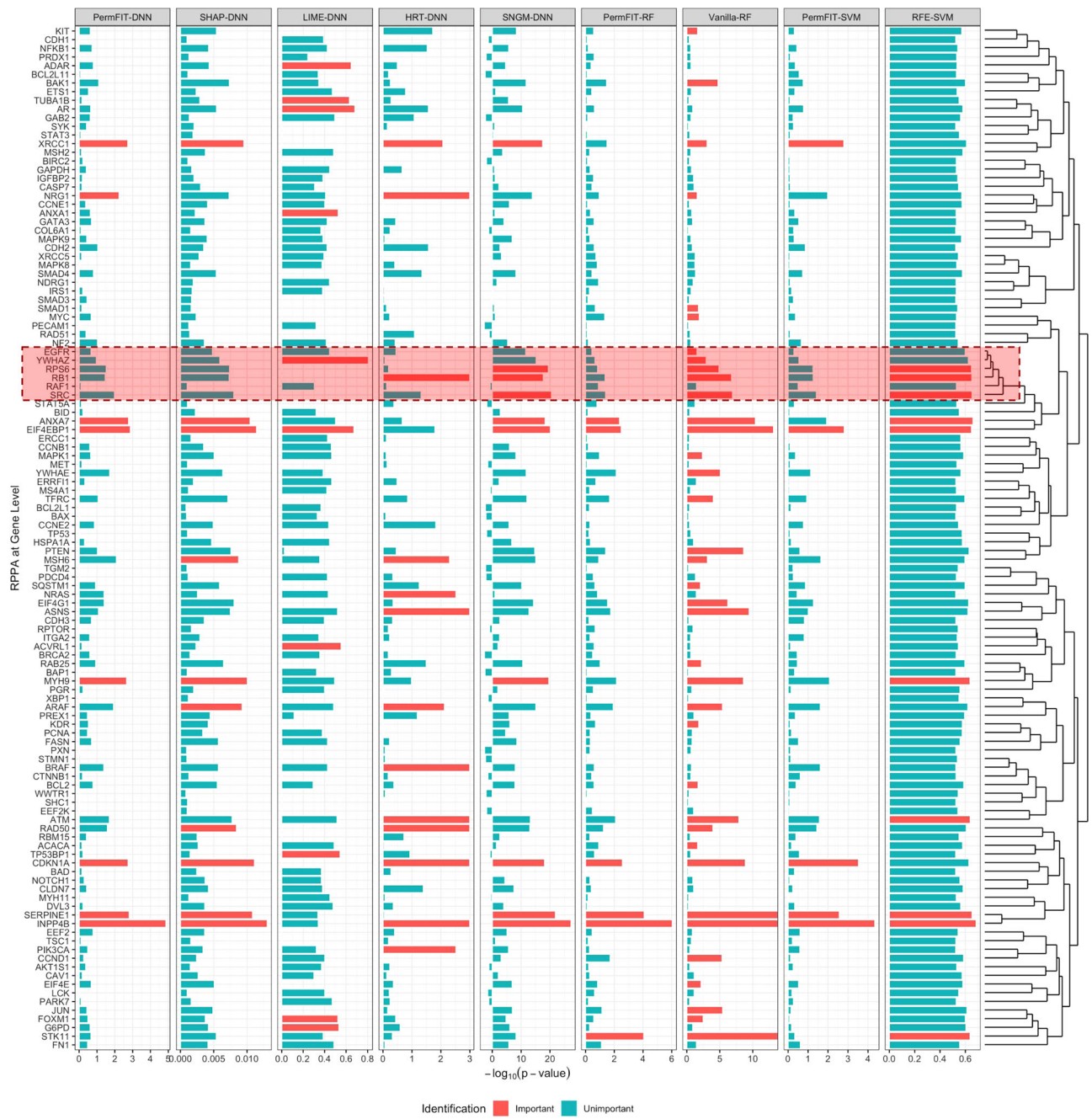

**Fig. 2 Negative log₁₀p values for TCGA kidney cancer data.** Important features selected by each method is marked in red. Since SHAP-DNN, LIME-DNN, SNGM-DNN, and RFE-SVM do not produce a p value, its importance is presented instead, and 10 features with top importance scores are marked. The highly correlated features (see details from the dendrogram on the right) selected by RFE-SVM, Vanilla-RF, and SNGM-DNN, but not by PermFIT methods, are highlighted. Source data are provided as a Source Data file.

However, six proteins, SRC, RAF1, RB1, RPS6, YWHAZ, and EGFR, are highly correlated and clustered together by hierarchical clustering in Fig. 2. Among them, EGFR, YWHAZ, RPS6, RB1, and SRC are identified by Vanilla-RF, and RPS6, RB1 and SRC are selected by RFE-SVM, while none of these biomarkers are selected by any PermFIT procedures. According to our observations in simulation studies, both Vanilla-RF and RFE-SVM tend to identify false positive biomarkers in the presence of high correlation among features, casting some doubts on the validity of their biomarker selection results. In addition, LIME-DNN identifies a very different set of important biomarkers compared to SHAP-DNN, HRT-DNN, SNGM-DNN, and PermFIT-DNN.

Since the underlying genetic truth is unknown, we alternatively use the model performance improvement estimated via 5-fold cross-validation, randomly repeated for 100 times (see Fig. 3a, b; Supplementary Table 2) as a surrogate measure for evaluating the relative quality of the selected features. Similar to the simulation study, we set the feature inclusion criteria on p values smaller than 0.1 for HRT-DNN, Vanilla-RF, and PermFIT methods, and top 20 features for SHAP-DNN, LIME-DNN, SNGM-DNN, and RFE-SVM. PermFIT-RF improves the accuracy from 0.694 to 0.732 on average, while Vanilla-RF only improves to 0.713. Moreover, PermFIT-SVM elevates the accuracy from 0.69 to 0.744, outperforming RFE-SVM (0.709). Similar to the simulation

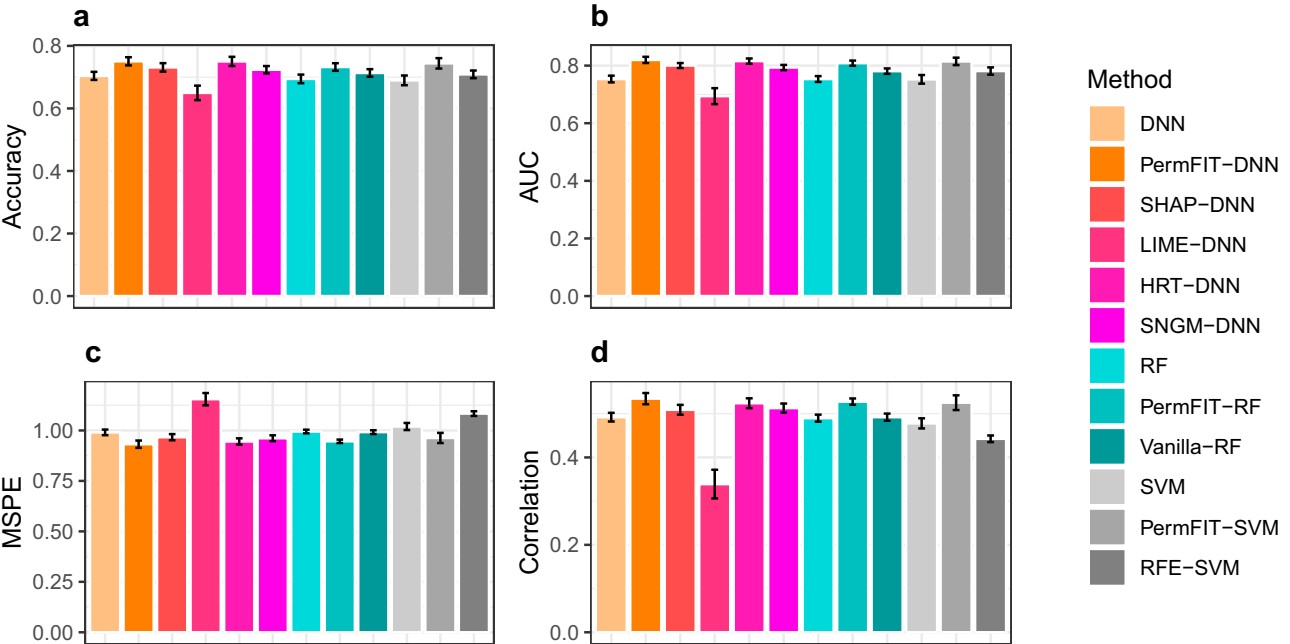

**Fig. 3 Model performance improvement from feature selection. a**, **b** Fivefold cross-validated prediction accuracy and AUC for TCGA kidney cancer data. **c**, **d** Fivefold cross-validated MSPE and Pearson correlation (between true outcome and prediction) for HITChip Atlas data. Fivefold cross-validation evaluation is randomly repeated for 100 times. Data are presented as mean values ± s.d. Source data are provided as a Source Data file.

results, PermFIT-DNN and HRT-DNN achieve the highest accuracy (0.751 and 0.750, respectively), better than those from SHAP-DNN (0.731), LIME-DNN (0.650) and SNGM-DNN (0.723). The same conclusion is further confirmed by area under the ROC curve (AUC) results. In summary, it is evident that PermFIT procedures consistently perform more efficient and accurate feature selection across various machine learning frameworks.

On the identified biomarkers, four genes—*CDKN1A*, *EIF4EBP1*, *INPP4B*, and *SERPINE1*—are claimed by all the three PermFIT methods to be significantly associated with the survival status. Interestingly, all four genes have been reported to be cancer related. Especially, *INPP4B*, identified as the most significant biomarker by all the three methods (*p* value = $1.3E - 05$ by PermFIT-DNN, $9.1E - 07$ by PermFIT-RF, and $4.5E - 05$ by PermFIT-SVM), encodes inositol polyphosphate-4-phosphatase, type II, a dual specificity phosphatase. Low *INPP4B* is recently reported to be associated with shorter survival in kidney clear cell, liver hepatocellular, and bladder urotheleal carcinomas, and with long survival in pancreatic adenocarcinoma[29]. It is also related to acute myeloid leukemia, breast cancer and bladder cancer[30–32]. *SERPINE1* encodes plasminogen activator inhibitor-1, which plays an important role in various diseases, in particular, kidney pathology and renal cell cancer[33–35]. In addition, the *CDKN1A* encoded protein, CDK-interacting protein 1, was reported as a prognostic marker for renal cell cancer[36], and has an effect on kidney cancer cell death[37] as well as kidney cancer survival[38]. Similarly, *EIF4EBP1* affects disease progression in renal cell carcinoma[39].

Moreover, the DNA repair protein *XRCC1*, identified by PermFIT-DNN and PermFIT-SVM, is shown to be associated with bladder cancer[40]. *ANXA7*, identified by PermFIT-DNN and PermFIT-RF, is reported to be associated with prostate cancer and breast cancer[41,42], and its encoded protein has an impact on prostate cancer and breast cancer[43,44]. Furthermore, *MYH9* and *NRG1* are identified by PermFIT-DNN. Myosin-9, encoded by *MYH9*, has been discussed for its role as a tumor suppressor[45], and *NRG1* is also reported to be related to multiple cancer

types[46,47]. Last, PermFIT-RF identifies a novel gene, *STK11* whose role in kidney cancer is unknown, however, it has been reported that inactivation of *STK11* in lung adenocarcinomas is a common event[48].

**HITChip atlas data**. In the HITChip Atlas study, the data was collected from 1006 adults in 15 western countries[49] by using the HITChip, and it is publicly available in R library "microbiome"[50]. Besides demographic and clinical variables, the HITChip Atlas data includes microbiome measurements from 130 taxonomic groups summarized at the genus level, which covered major types of human intestinal microbiota bacterial diversity. Many of the 130 taxonomic groups are highly correlated, which is reflected in the correlation heat map and the hierarchical clustering dendrogram (see Fig. 4 and Supplementary Fig. 4). We then investigated the importance of demographic factors, including gender and nationality, together with 129 microbial genus (1 was removed due to the use of compositional values), in predicting the baseline BMI level. Our analysis includes 900 subjects with BMI measurements.

The feature selection and biomarker identification criteria remain the same as those in the TCGA example. The improvement of the model performance from variable selection is presented in Fig. 3c, d and Supplementary Table 2. Besides the MSPE, we report the Pearson correlation between the predicted and the true values. We notice that high correlation among microbiome features leads to large inflation in importance scores from Vanilla-RF, corresponding to high false positive rate. As a result, it fails to improve the model performance and the reduced model with selected features from RFE-SVM performs worse than the full model. Again, this is likely due to the fact that highly correlated biomarkers are falsely selected by RFE-SVM. For instance, *Streptococcus mitis et rel*, *Streptococcus bovis et rel* and *Streptococcus intermedius et rel*, highly correlated to each other, are among top 10 biomarkers identified by RFE-SVM. In contrast, PermFIT yields the most remarkable improvements in all these models, reflected in both MSPE and correlation.

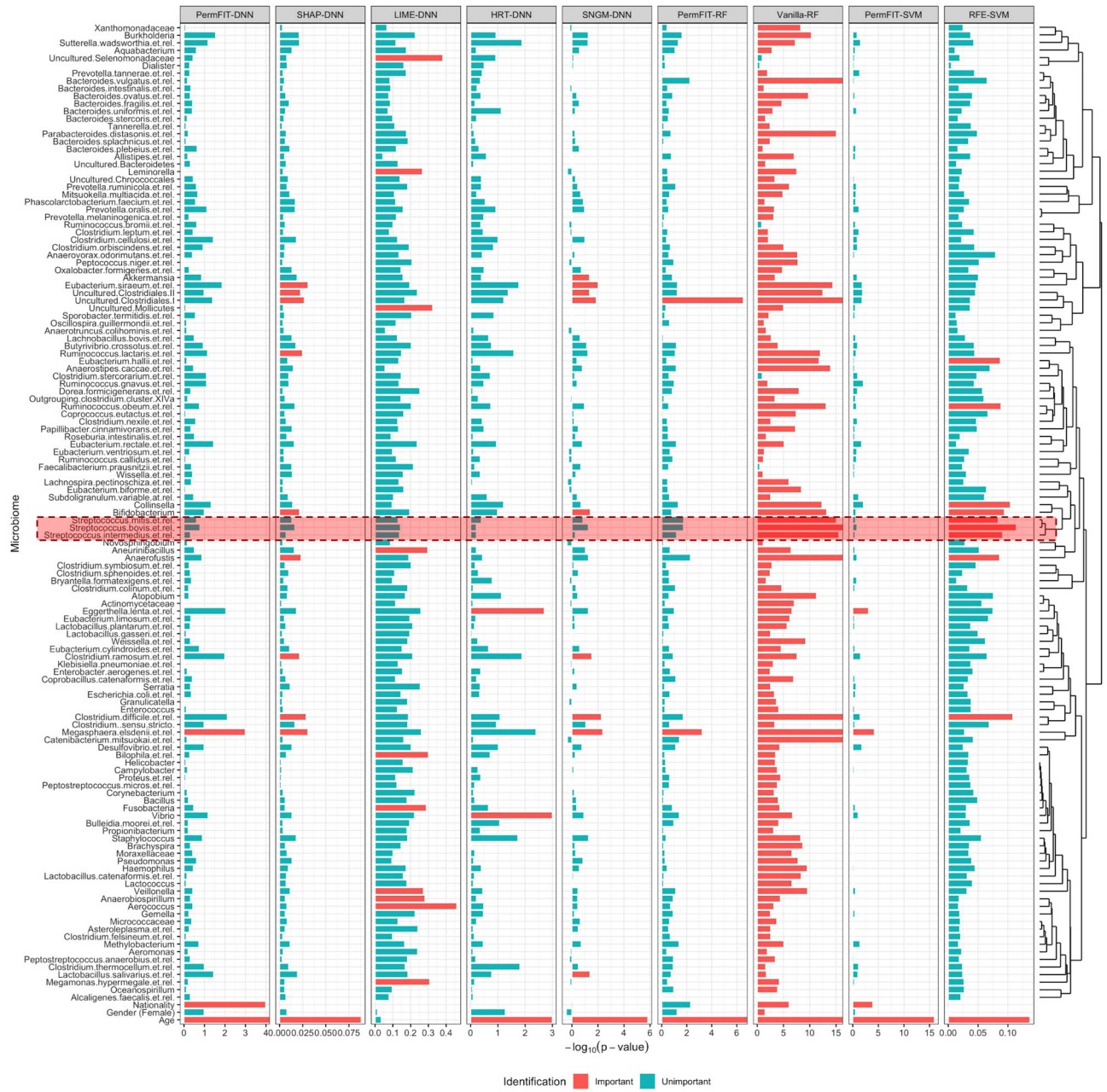

**Fig. 4 Negative log$_{10}$p values for HITChip Atlas data.** Important features selected by each method is marked in red. Since SHAP-DNN, LIME-DNN, SNGM-DNN, and RFE-SVM do not produce a p-value, its importance is presented instead, and 10 features with top importance scores are marked. The highly correlated features (see details from the dendrogram on the right) selected by RFE-SVM and Vanilla-RF, but not by PermFIT methods, are highlighted. Source data are provided as a Source Data file.

Figure 4 and Supplementary Fig. 5 show the negative log$_{10}$(p-value)s and the importance scores estimated from each method. Among all the features, as expected, age is identified as the most significant factor. The nationality is also selected by PermFIT-DNN and PermFIT-SVM. Among all the microbiome features, *Megasphaera elsdenii* is identified by all the PermFIT methods. *M. elsdenii* is shown by prior studies as one of the ruminal and intestinal lacate- and sugar-fermenting species[51]. *M. elsdenii* is also reported to massively reside in patients with an increase in BMI after bariatric surgery[52]. In addition, *Eggerthella lenta* is identified by PermFIT-SVM. *E. lenta* is not well-studied, but its potential role as an emerging pathogen has been increasingly recognized in years[53]. Lastly, uncultured clostridiales is identified by PermFIT-RF.

## Discussion

Complex machine learning models are difficult to distinguish the contribution of individual input features, though they enjoy the more robustness and flexibility in modeling complex human diseases as compared with parametric models. In this paper, we introduce PermFIT, a computationally efficient permutation-based feature importance test with applicability to various machine learning models such as DNN, RF, and SVM, to identify important features. Also, as demonstrated by the applications to TCGA kidney cancer data and HITChip Atlas BMI data, Perm-FIT procedures further show the superior performance over all the other competitors of concern in the paper, which severely suffer from false positive or negative findings, leading to inferior prediction performance with top selected features. In contrast,

feature selection via PermFIT procedures remarkably improves the performance of these predictive models. However, it is worth pointing out that the prediction improvement of PermFIT is restricted to the capability of each machine learning model framework. For example, RF is relatively inefficient in modeling interaction terms, thus the performance of PermFIT-RF may be limited for complex traits with strong gene-gene interactions. Overall, PermFIT coupled with DNN consistently shows superior empirical performance.

The proposed analytical tool, PermFIT, is computationally efficient and has broad applicability in addressing real-world problems. It can be implemented and incorporated into a variety of machine learning models with different types of outcomes, and without the need of model refitting. PermFIT provides researchers a useful tool for deciphering complex genetic architecture and disease etiologies of complex traits.

## Methods

**PermFIT.** We start the case with a continuous outcome. Let $X \in \mathcal{X}$, $Y \in \mathcal{Y}$, where $X = (X_1, \ldots, X_p)$ is a $p$-dimensional covariate vector, the observation of the outcome variable $Y$ is a continuous scalar, $E(Y|X) = \mu(X)$, $\mu(\cdot)$ is an unknown mapping from $\mathcal{X}$ to $\mathcal{Y}$, the residues $\epsilon = Y - \mu(X)$ is independent of $X$ and $0 < \sigma^2 = E(\epsilon^2) < \infty$.

We define the feature importance score $M_j$ for $X_j$, the $j^{\text{th}}$ feature in $X(j = 1, \ldots, p)$, as the expected squared difference between $\mu(X)$ and $\mu(X^{(j)})$, where $X^{(j)} = (X_1, \ldots, X_{j-1}, X_{j'}, X_{j+1}, \ldots, X_p)$ is equal to $X$ with the $j^{\text{th}}$ covariate replaced by a random vector $X_{j'}$ whose elements are independently drawn from the distribution of $X_j$. The importance score $M_j$ can be expressed as,

$$M_j = E_{X, X_{j'}} \left[ \mu(X) - \mu(X^{(j)}) \right]^2. \quad (3)$$

Assuming $X$ does not have redudant features, $M_j$ is zero only when $\mu(X) \equiv \mu(X^{(j)})$ on $\mathcal{X}$, implying that the $j^{\text{th}}$ element of $X$ does not have any impact on $\mu(X)$; and is non-zero otherwise.

To obtain a clear understanding of $M_j$, we take the linear model as an example where $\mu(X) = X\beta + \beta_0$, with $\beta = (\beta_1, \ldots, \beta_p)$ consisting of $p$ parameters. Under the linear assumption, (3) becomes:

$$M_j = E_{X, X_{j'}} (X_j - X_{j'})^2 \beta_j^2 = 2\beta_j^2 \text{Var}(X_j). \quad (4)$$

Here, (4) is proportional to the squared standardized coefficient, which has been recognized as a popular measure of variable importance in multiple linear regression. Furthermore, $M_j$ can be simply decomposed as follows:

$$M_j = E_{X, X_{j'}} \left[ Y - \mu(X^{(j)}) \right]^2 - E_X [Y - \mu(X)]^2. \quad (5)$$

Ideally, given the true form of $\mu(\cdot)$, from (5), $M_j$ could be estimated through permutation. Let $(Y_i, X_{i1}, \ldots, X_{ip})$, $(i = 1, \ldots, N)$ be $N$ independent observations drawn from the distribution of $(Y, X_1, \ldots, X_p)$. A permutation on one covariate $X_j = (X_{1j}, \ldots, X_{Nj})$ is to randomly sample the elements in $X_j$ without replacement to generate a permuted version of $X' = (X_{s_1 j}, \ldots, X_{s_N j})$. The empirical permutation importance score is then,

$$M_j^{(P)} = \frac{1}{N} \sum_{i=1}^{N} \left[ \left\{ Y_i - \mu(X_{i \cdot}^{(j)}) \right\}^2 - \left\{ Y_i - \mu(X_{i \cdot}) \right\}^2 \right], \quad (6)$$

where $X_{i \cdot} = (X_{i1}, \ldots, X_{ip})$ and $X_{i \cdot}^{(j)} = (X_{i1}, \ldots, X_{i,j-1}, X_{s_i, j}, X_{i,j+1}, \ldots, X_{ip})$. Let $M_j^{(P)} = \frac{1}{N} \sum_{i=1}^{N} M_{ij}^{(P)}$, where $M_{ij}^{(P)} = \left\{ Y_i - \mu(X_{i \cdot}^{(j)}) \right\}^2 - \left\{ Y_i - \mu(X_{i \cdot}) \right\}^2$, then

$$E\left[ M_j^{(P)} \right] = E\left[ M_{ij}^{(P)} \right] = \frac{N-1}{N} M_j. \quad (7)$$

When $N$ is large, $M_j$ could be well approximated by $M_j^{(P)}$. Besides, $\text{Var}\left[ M_j^{(P)} \right] \approx \frac{1}{N} \text{Var}\left[ M_{ij}^{(P)} \right]$ which can be approximated by the empirical variance of $M_{ij}^{(P)}$.

Let $\widehat{\mu}(\cdot)$ be the fitted function approximator to $\mu(\cdot)$, according to (6), we propose to estimate $M_j^P$ by

$$\widehat{M}_j^{(P)} = \frac{1}{N} \sum_{i=1}^{N} \left[ \left\{ Y_i - \widehat{\mu}(X_{i \cdot}^{(j)}) \right\}^2 - \left\{ Y_i - \widehat{\mu}(X_{i \cdot}) \right\}^2 \right]. \quad (8)$$

If feature $X_j$ is not associated with $Y$, then $\mu(X_{i \cdot}^{(j)}) = \mu(X_{i \cdot})$ with corresponding $M_j^{(P)} = 0$, and Eq. (8) becomes,

$$\widehat{M}_j^{(P)} = \frac{1}{N} \sum_{i=1}^{N} \left[ \left\{ \mu(X_{i \cdot}) - \widehat{\mu}(X_{i \cdot}^{(j)}) \right\}^2 - \left\{ \mu(X_{i \cdot}) - \widehat{\mu}(X_{i \cdot}) \right\}^2 + 2\epsilon_i \left\{ \widehat{\mu}(X_{i \cdot}) - \widehat{\mu}(X_{i \cdot}^{(j)}) \right\} \right]. \quad (9)$$

With the universal consistency, the three terms are expected to converge to zero as $N$ goes to infinity. However, for data with a finite sample size, the model $\widehat{\mu}(\cdot)$ may become overfit, leading to $\left\{ \mu(X_{i \cdot}) - \widehat{\mu}(X_{i \cdot}^{(j)}) \right\}^2 > \left\{ \mu(X_{i \cdot}) - \widehat{\mu}(X_{i \cdot}) \right\}^2$ in estimating $M_j^{(P)}$. To overcome the bias issue, we employ cross-fitting strategy to separate the input data as the training and validation sets, with one set being utilized to obtain $\widehat{\mu}(\cdot)$ and the other to estimate $\widehat{M}_j^{(P)}$. Let $\widehat{\mu}_T(\cdot)$ be the estimate of $\mu(\cdot)$ from the training set, and $\mathcal{D}_V = \{Y_i, X_{i \cdot}\}_{i=1}^{N_V}$ be the validation set,

$$\widehat{M}_j^{(P)} = \frac{1}{N_V} \sum_{i=1}^{N_V} \left[ \left\{ Y_i - \widehat{\mu}_T(X_{i \cdot}^{(j)}) \right\}^2 - \left\{ Y_i - \widehat{\mu}_T(X_{i \cdot}) \right\}^2 \right], \quad (10)$$

$$\widehat{\text{Var}}\left[ \widehat{M}_j^{(P)} \right] = \frac{1}{N_V} \sum_{i=1}^{N_V} \left[ \left\{ Y_i - \widehat{\mu}_T(X_{i \cdot}^{(j)}) \right\}^2 - \left\{ Y_i - \widehat{\mu}_T(X_{i \cdot}) \right\}^2 - \widehat{M}_j^{(P)} \right]^2. \quad (11)$$

The one-sided $p$ value can be obtained by assuming normality. To increase the power of important feature identification, $K$-fold cross-fitting can be adopted. Here, we randomly divide the data into $K$ folds, denoted as $V_1, \ldots, V_K$. For each of $V_k$, $k = 1, \ldots, K$, $\overline{V}_k$ denote the complementary set of $V_k$, which is used to fit the model $\widehat{\mu}_k(\cdot)$. Then

$$\widehat{M}_{ij}^{(P,CV)} = \sum_{k=1}^{K} I(i \in V_k) \left[ \left\{ Y_i - \widehat{\mu}_T(X_{i \cdot}^{(j)}) \right\}^2 - \left\{ Y_i - \widehat{\mu}_k(X_{i \cdot}) \right\}^2 \right], \quad (12)$$

$$\widehat{M}_j^{(P,CV)} = \frac{1}{N} \sum_{i=1}^{N} \widehat{M}_{ij}^{(P,CV)}, \quad (13)$$

$$\widehat{\text{Var}}\left[ \widehat{M}_j^{(P,CV)} \right] = \frac{1}{N} \sum_{k=1}^{K} \sum_{i \in V_k} \left[ \left\{ Y_i - \widehat{\mu}_T(X_{i \cdot}^{(j)}) \right\}^2 - \left\{ Y_i - \widehat{\mu}_k(X_{i \cdot}) \right\}^2 - \widehat{M}_j^{(P,CV)} \right]^2. \quad (14)$$

The algorithm of PermFIT with cross-fitting is illustrated in Algorithm 1.

## Algorithm 1

Algorithms for PermFIT
1: Randomly divide the data into $K$ folds.
2: **for** $k = 1$ **to** $K$ **do**
3:    Denote the data in $k^{\text{th}}$ fold as $V_k$ and the rest of the data as $\overline{V}_k$.
4:    Build the machine learning model with $\overline{V}_k$, denoted as $\widehat{\mu}_k(\cdot)$.
5:    **for** $j = 1$ **to** $p$ **do**
6:       Calculate $\widehat{M}_{ij}^{(P,CV)}$ for subjects in $\mathcal{D}_k$.
7:    **end for**
8: **end for**
9: **for** $j = 1$ **to** $p$ **do**
10:    Calculate $\widehat{M}_j^{(P,CV)}$ and estimate $\widehat{\text{Var}}\left[ \widehat{M}_j^{(P,CV)} \right]$. Calculate p-value by assuming nomality.
11: **end for**

**Binary outcome.** For binary outcome $Y \in \{0, 1\}$, we have $\mu(X) = E(Y|X) = \Pr(Y = 1|X)$ and define $M_j$ as the expectation of binomial deviance,

$$M_j = E_{X, X_{j'}} \left[ Y \log \left( \frac{\mu(X)}{\mu(X^{(j)})} \right) + (1 - Y) \log \left( \frac{1 - \mu(X)}{1 - \mu(X^{(j)})} \right) \right]. \quad (15)$$

The empirical estimate of $M_j$ can be similarly obtained by plugging in the estimate of $\mu(X^{(j)})$ and $\mu(X)$ as in the continuous data scenario.

**DNN with bootstrap aggregating.** In this paper, we use feedforward and fully-connected deep neural networks (DNNs) to approximate function $\mu(\cdot)$. The DNN model contains $L$ hidden layers of $(n_1, \ldots, n_L)$ hidden nodes that transform the initial input covariates $X$ to the estimation of the continuous outcome $Y$. Let $\theta$ denote all the parameters in the DNN model, we essentially have the fitted DNN, $\widehat{\mu}(X; \theta)$, by minimizing the empirical risk function,

$$\arg \min_{\theta} \frac{1}{N} \sum_{i=1}^{N} \ell\{Y_i, \mu(X_i; \theta)\} + \lambda \Omega(\theta), \quad (16)$$

where $\ell(\cdot, \cdot)$ is the loss function dependent on the outcome type, $\Omega(\theta)$ is a penalty on $\theta$ and $\lambda$ is a hyperparameter that controls the degree of regularization, via mini-batch stochastic gradient descent algorithm and Adam[54] to adjust the learning rate.

To increase the robustness and accuracy of DNNs, bootstrap aggregating (bagging) is applied[55]. Besides, due to the randomness of initial parameters, some DNNs may not converge to a stable solution, hence, perform poorly. In neural network ensembles, it is argued that "many could be better than all", meaning that using a subset of bagged DNNs that well fit the data could be better than using all bagged DNNs[25,56]. Therefore, we adopt the scoring system to select the optimal subset of DNNs in the bagging procedure, following Mi et al.[25]. DNN with bagging has been implemented in the R package "deepTL" (available at https://github.com/SkadiEye/deepTL)[57]. According to Mi et al.[25], for all the reported numerical analysis in this paper, we set bagging size to 100, batch size to 50, the

number of hidden layers to 4 with 50, 40, 30, 20 hidden nodes at each layer subsequently, penalty weight $\lambda$ to $1E - 4$, and reclified linear units as the activation function.

**SHAP, LIME, SNGM, and HRT**. Shapley and LIME values are calculated using R package "iml". Feature importance scores of SHAP and LIME are defined as mean absolute values of Shapley and LIME values, respectively. HRT is implemented in R. SNGM-DNN is implemented in R, following Xing et al.[21]. To be consistent with other approaches without providing p values, implementation of SNGM-DNN also focuses on selecting top features. In SHAP-DNN, LIME-DNN, SNGM-DNN and HRT-DNN, the above-described bagged DNNs are applied.

**RF and SVM**. RF is implemented via R package "randomForest". Vanilla-RF importance and its standard error is generated from the "randomForest" function with 1000 trees. SVM is implemented through R package "e1071" with Radial kernels used. The hyper-parameters in SVMs are searched via fivefold cross-validation. RFE-SVM was implemented with the function "rfe" in R package "caret".

**The simulation study and real data applications**. In the simulation studies, PermFIT is performed by randomly splitting the samples into training (80%) and validation (20%) sets, and the importance score is estimated via (10) and (11). In real applications, HRT and PermFIT are conducted with 5-fold cross-fitting through (13) and (14). To eliminate the impact from the randomness of cross-fitting and other random factors in model fitting, we repeat each method 100 times and report the mean and standard deviation of MSPE, Pearson correlation, AUC or accuracy, and the median of the importance scores and $p$ values. Features presented in figures are ordered by hierarchical clustering, which is implemented in "hclust" function in R package "stats", where the dissimilarity is set to one minus the Pearson correlation.

For the TCGA kidney cancer application, RPPAs at gene level are analyzed. We first remove the proteins that are not common across all three TCGA datasets (KIRC, KIRP, and KICH). In addition, we remove the proteins with perfect multicollinearity, after which 118 are kept for further analysis.

For the HITChip Atlas data, the BMI level was originally grouped into six groups: underweight, lean, overweight, obese, severeobese and morbidobese, which we transform into numerical levels from 1 to 6 in our analysis. Total 900 subjects are left for the analysis after subjects with missing BMI are excluded. Missing information on nationality is grouped into a new group named "Unknown". Missing values in the microbiome data are simply imputed with the median values across all samples. The analysis on the microbiome data is based on the compositional values but we remove the cell proportion from the last group due to the sum to 1 constraint on the compositional values, after which a log-transformation is applied to the remaining compositions.

**Reporting summary**. Further information on research design is available in the Nature Research Reporting Summary linked to this article.

## Data availability

The TCGA datasets are available at the LinkedOmics website (http://linkedomics.org), among which three studies, KIRC, KICH, and KIRP, are used (dbGaP Study Accession: phs000178). The HITChip Atlas data is available in R package "microbiome" (https://microbiome.github.io/). We provide final datasets used in our analysis (https://github.com/SkadiEye/deepTL/tree/master/permfit/code/cleaned-dat.RDS). Source data are provided with this paper.

## Code availability

PermFIT is implemented in our R package "deepTL" (https://github.com/SkadiEye/deepTL)[57]. We also provide source code for replicating the simulation studies and real data applications (https://github.com/SkadiEye/deepTL/tree/master/permfit/code).

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

## Acknowledgements

The work was partially supported by the National Institute of Health Grants R01AI143886, R01CA219896, CCSG P30 CA013696, and P30 ES010126.

## Author contributions

X.M. implemented the algorithms in "deepTL" for the proposed method and performed numerical analyses. All authors contributed to the methodology development and writing the manuscript.

## Competing interests

The authors declare no competing interests.
