## [Peer Review File · Nature Communications]

Reviewers' Comments:

Reviewer #1:

Remarks to the Author:

Summary:

This manuscript proposes a model-agnostic feature importance measure, named PermFIT, to evaluate the feature importance for the purpose of interpreting individual feature in various black-box frameworks, including deep neural networks (DNN), random forests (RF), and support vector machines (SVM). Specifically, PermFIT first approximates the function that maps features to the outcome, based on which, PermFIT then evaluates the importance score of each feature, defined as the expected increase of prediction errors due to the permutation of the feature.

Results are provided on both simulated and real dataset. For real data analysis, PermFIT is applied two benchmark datasets: the Reverse Phase Protein Arrays (RPPAs) data from three kidney cancer studies in The Cancer Genomic Atlas (TCGA) and the HITChip Atlas microbiome data regarding body mass index (BMI). PermFIT demonstrates its practical usage in identifying important biomarkers and boosting performance of black-box predictive models.

Detailed comments:

Strengths

- (1) The paper aims to tackle an interesting problem: model-agnostic explanation by feature selection.
- (2) The paper is well-written and easy to follow.

Weaknesses

(1) The literature review is not comprehensive. Though the manuscript listed a few such as knockoff models, permutation-based feature importance, etc. several widely-used model-agnostic explanation methods are missing, such as LIME [1] and SHAP [2]. In particular, the philosophy of LIME is very similar to the proposed method.

(2) Several claims in the manuscript are not flawless, such as:

- (a). In page 3, "The overall disadvantage of CRT, HRT and model-X knockoff is that they all depend on the assumption of a known covariance structure". However, the extension of model-X knockoff, KnockoffGAN [3], does not suffer the mentioned assumption.
- (b). In page 13, "The M_j is zero only when ..., implying that the j -th element of X does not have any impact on $\mu(X)$ " is only true when X does not have redundant features.

(3) In the experiments, at least four baselines are missing:

- (a) holdout randomization test;
- (b) knockoff+DNN method: DeepPINK;
- (c) model-agnostic explanation method: LINE;
- (d) model-agnostic explanation method: SHAP. Without comparing against these four methods, it is hard to tell whether PermFIT is among the state-of-the-arts.

(4) The Figure 1b, the mean square prediction error (MSPE) is suspicious because one would expect the MSPE produced by DNN, SVM, and RF are of similar range, such as Figure 3c.

(5) PermFIT is claimed computationally efficient, however, it's hard to believe because bootstrap aggregation is involved. Running time is expected to be provided.

Reproducibility:

Code is available on GitHub, but I didn't try to reproduce the results by myself

Expertise:

Expert

Confidence:
Expert

Reference:

- [1] Ribeiro, Marco Tulio, Sameer Singh, and Carlos Guestrin. "" Why should i trust you?" Explaining the predictions of any classifier." Proceedings of the 22nd ACM SIGKDD international conference on knowledge discovery and data mining. 2016.
- [2] Lundberg, Scott M., and Su-In Lee. "A unified approach to interpreting model predictions." Advances in neural information processing systems. 2017.
- [3] Jordon, James, Jinsung Yoon, and Mihaela van der Schaar. "KnockoffGAN: Generating knockoffs for feature selection using generative adversarial networks." (2018).

Reviewer #2:

Remarks to the Author:

Major Comments:

I strongly disagree with denoting the machine learning models as black box models. We may hear people refer to modern machine learning systems as "black boxes". However, machine learning models, including deep learning systems, are not black boxes; even the development of such a system need not be a black box. I agree that both of these things are complex, and not necessarily well understood. That is why sometimes people tend to call them black box which is quite misleading. I request the authors to avoid using the term 'black box' and call those as 'machine learning models'.

The authors first build up a machine learning model (RF, SVM, or DNN) using all the input features. Then they apply the PermFIT algorithm to find out the top K important input features based on the coefficient terms or input feature weights using equation (9) and (10). This is good that besides prediction tasks they also propose a technique for finding out the important input features which can be considered as biomarkers. Later they refit the model using those top K input features only which is said to improve the prediction accuracy. My concerns are below:

1. Let us denote the original model or network as D . Later when they are using only top K features, do they bring any changes to D ? For instance, in the case of DNN, do they change the number of layers or neurons? It seems like they only change the number of inputs in the first layer. In that case the model might overfit since it was optimized for too many inputs before.

2. If they have made the size of the original model D smaller and denote the new model as D' , then did they apply D' to the original problem with all the input features? And compare the prediction accuracy? Because D' can give better predictions even with all the input features by automatically emphasizing on those Top K features, thus finding the important features by itself.

My point is, in the case of DNN, people tend to experiment with network size to find out the optimal size. If this is done appropriately then DNN is supposed to find out the important features itself. This is one of the core characteristics of DNN. Dropout and other techniques are also available to avoid overfit. Did the authors apply those techniques? It might happen that original model D was overfit or underfit and PermFIT is helping to find out the optimal model size instead of running multiple random experiments to determine the optimal size. Which is appreciating. But I am not completely convinced that DNN is unable to achieve better performance without PermFIT. Maybe more thorough experiments are necessary to establish their claim.

Minor Comments:

In the algorithm, I see the line 10 uses equations 9 and 10. But no equation is provided for line 6.

Even if their claim is correct that PermFit improves prediction accuracy for machine learning algorithms, it might not be applicable in general context. For example, image or vision based problems might not benefit from PermFit. So the last sentence in abstract might be rephrased so that people are not confused.

Dear Reviewers,

We are grateful for your constructive suggestions and comments, which have helped us improve the manuscript greatly. Below please find our point-by-point responses to the raised suggestions and comments (in italics).

Responses to comments from Reviewer 1

This manuscript proposes a model-agnostic feature importance measure, named PermFIT, to evaluate the feature importance for the purpose of interpreting individual feature in various black-box frameworks, including deep neural networks (DNN), random forests (RF), and support vector machines (SVM). PermFIT demonstrates its practical usage in identifying important biomarkers and boosting performance of black-box predictive models.

Detailed comments:

Strengths

- (1) The paper aims to tackle an interesting problem: model-agnostic explanation by feature selection.*
- (2) The paper is well-written and easy to follow.*

Response: Thank you for your positive comments.

Weaknesses

- (1) The literature review is not comprehensive. Though the manuscript listed a few such as knockoff models, permutation-based feature importance, etc. several widely-used model-agnostic explanation methods are missing, such as LIME [1] and SHAP [2]. In particular, the philosophy of LIME is very similar to the proposed method.*

Response: Thank you for the suggestions. We have added LIME and SHAP methods for the comparison in the revised paper.

- (2) Several claims in the manuscript are not flawless, such as: (a). In page 3, “The overall disadvantage of CRT, HRT and model-X knockoff is that they all depend on the assumption of a known covariance structure”. However, the extension of model-X knockoff, KnockoffGAN [3], does not suffer the mentioned assumption.*

Response: Thank you for pointing this out. We have now included KnockoffGAN in Introduction section on Page 3 to improve the literature review coverage. We note that, even though KnockoffGAN does not depend on a known covariance structure, it is difficult to train adversarial networks and requires more tuning.

- (b). In page 13, “The M_j is zero only when ..., implying that the j -th element of X does not have any impact on $\mu(X)$ ” is only true when X does not have redundant features.*

Response: We appreciate this point. We have now modified the corresponding sentence on Page 14 of the revised paper to make the statement more appropriate.

- (3) In the experiments, at least four baselines are missing:*

(a) holdout randomization test; (b) knockoff+DNN method: DeepPINK; (c) model-agnostic explanation method: LIME; (d) model-agnostic explanation method: SHAP. Without comparing against these four methods, it is hard to tell whether PermFIT is among the state-of-the-arts.

Response: Thank you for your suggestions. We have now applied holdout randomization test (HRT), LIME and SHAP to the DNN models. However, we did not include DeepPINK since it is developed for DNN models with a special designed structure, which is not applicable to our empirical investigation. In addition, DeepPINK can not be generalized to other machine learning methods. We have also updated the tables and figures regarding simulation and real data results corresponding to these methods. In light of your suggestion, we added two more challenging simulation scenarios with smaller sample sizes ($n = 300, 500$) and a larger number of the input features ($p = 200$), to further investigate the performance of each method. The results are presented in Table 2 on Page 28. In summary, the performance of HRT largely depends on the estimation accuracy of the covariance matrix of the input features, and may fail to preserve type-I error, while both LIME and SHAP have low power, especially for data with small sample sizes.

(4) The Figure 1b, the mean square prediction error (MSPE) is suspicious because one would expect the MSPE produced by DNN, SVM, and RF are of similar range, such as Figure 3c.

Response: Thanks for raising this concern. In Figure 1b, MSPE of RF is indeed inferior to that of DNN. A potential explanation is that RF is not as efficient as the other approaches in modeling interaction terms. The inferior performance of SVM is likely because of its incapability of identifying truly important features in the high-dimensional data ($p = 100$), which is evident by the observation that the SVM model using the features selected by PermFIT has dramatically reduced MSPE. The real application presented in Figure 3c does show that the differences among these methods is smaller, suggesting that the underlying (unknown) data structure in the real application might deviate from our simulation scenarios.

(5) PermFIT is claimed computationally efficient, however, it's hard to believe because bootstrap aggregation is involved. Running time is expected to be provided.

Response: Thank you for raising the concern. We have now reported the runtime in Table S3 in the supplement material. In addition, we would like to clarify that we claim the proposed method is computationally efficient in the sense that PermFIT methods do not need to refit models for each permuted variable.

Responses to comments from Reviewer 2

I strongly disagree with denoting the machine learning models as black box models. We may hear people refer to modern machine learning systems as "black boxes". However, machine learning models, including deep learning systems, are not black boxes; even the development of such a system need not be a black box. I agree that both of these things are complex, and not necessarily well understood. That is why sometimes people tend to call them black box which is quite misleading. I request the authors to avoid using the term 'black box' and call those as 'machine learning models'.

Response: Thank you very much for your valuable suggestions. We have replaced the phrase "black-box"

with "machine learning".

My concerns are below:

1. Let us denote the original model or network as D . Later when they are using only top K features, do they bring any changes to D ? For instance, in the case of DNN, do they change the number of layers or neurons? It seems like they only change the number of inputs in the first layer. In that case the model might overfit since it was optimized for too many inputs before.

2. If they have made the size of the original model D smaller and denote the new model as D' , then did they apply D' to the original problem with all the input features? And compare the prediction accuracy? Because D' can give better predictions even with all the input features by automatically emphasizing on those Top K features, thus finding the important features by itself.

My point is, in the case of DNN, people tend to experiment with network size to find out the optimal size. If this is done appropriately then DNN is supposed to find out the important features itself. This is one of the core characteristics of DNN. Dropout and other techniques are also available to avoid overfit. Did the authors apply those techniques? It might happen that original model D was overfit or underfit and PermFIT is helping to find out the optimal model size instead of running multiple random experiments to determine the optimal size. Which is appreciating. But I am not completely convinced that DNN is unable to achieve better performance without PermFIT. Maybe more thorough experiments are necessary to establish their claim.

Response: The above two concerns are highly related. On point 1, the model size of all DNNs (in terms of # of layers and # of nodes) is fixed across all scenarios (simulation studies and real analyses). On point 2, we totally agree that DNNs with the optimal size would achieve better prediction performances than those without the optimal size. However, in practice, the search of the optimal model size is not an easy task, and is often done in an ad-hoc fashion. Furthermore, DNNs with the optimal size can still lead to reduced prediction power if a large number of nuisance variables are included. Nonetheless, the paper is not on the selection of optimal model sizes, but on the selection of important features. Prediction is used to demonstrate the important utility of feature selection which we focus on in the paper, while DNNs with a reasonable model size is just a vehicle.

In light of your comments, we have included additional simulation results using DNNs with different number of hidden layers in Table S4 of the supplement material. Despite of the observed varying performances between DNNs with different numbers of hidden layers, the differences are minor compared to the improvement from PermFIT.

Minor Comments: In the algorithm, I see the line 10 uses equations 9 and 10. But no equation is provided for line 6.

Response: Thank you. We now added equation 9 which is used for line 6 of the algorithm. The equations that are used for line 10 are now re-numbered to 10 and 11.

Even if their claim is correct that PermFit improves prediction accuracy for machine learning algorithms,

it might not be applicable in general context. For example, image or vision based problems might not benefit from PermFit. So the last sentence in abstract might be rephrased so that people are not confused.

Response: Thank you very much for the suggestion. We agree and have modified the last sentence in the abstract to avoid the confusion.

Reviewers' Comments:

Reviewer #1:

Remarks to the Author:

Overall, the authors did a great job in addressing my comments. However, there are still some concerns remaining, listed as follows:

1. At least one knockoff method need to be compared. I agree that KnockoffGAN is complicated to train, but I am not convinced by the argument that DeepPINK is empirically inapplicable. In addition, there exists alternative knockoff methods without suffering the known covariance structure assumption, such as the Gaussian mirror method as its followup DNN-based extensions ("False Discovery Rate Control via Data Splitting").

2. I noticed that for SVM, the hyper-parameters are selected via cross-validation whereas for other methods the hyper-parameter selection is not mentioned. I am wondering if this is the reason why DNN-based method obtains such a good MSPE (because of overfitting?). More detailed investigation is needed.

Reviewer #2:

Remarks to the Author:

The authors have satisfactorily dealt with all of my concerns.

Responses to comments from Reviewer 1

1. At least one knockoff method need to be compared. I agree that KnockoffGAN is complicated to train, but I am not convinced by the argument that DeepPINK is empirically inapplicable. In addition, there exists alternative knockoff methods without suffering the known covariance structure assumption, such as the Gaussian mirror method as its followup DNN-based extensions (“False Discovery Rate Control via Data Splitting”).

Response: Thank you for your comments. Per your suggestion, we have added a Gaussian-mirror based method in comparison (i.e. simultaneous neural Gaussian mirror). We have also added a description of Gaussian mirror-based methods in the last paragraph of Introduction section on Page 3.

2. I noticed that for SVM, the hyper-parameters are selected via cross-validation whereas for other methods the hyper-parameter selection is not mentioned. I am wondering if this is the reason why DNN-based method obtains such a good MSPE (because of overfitting?). More detailed investigation is needed.

Response: Thanks for raising the concern. We used cross-validation in real data studies, and independent testing sets in simulation studies to evaluate MSPE. Therefore, in all our empirical investigations, the data used to evaluate MSPE are different from those used to train the models. Therefore, overfitting should not be a concern. Moreover, our DNN-based method did not fine-tune the hyper-parameters in both the simulation studies and real data analysis. Instead, a set of pre-selected number of layers and number of nodes were employed for all our analyses which actually less favors the DNN-based method. The consistently good performance of the DNN-based method somewhat suggests its robustness to the tuning

parameters. In Table S4 of Supplement, we also show that the network structure has negligible impact on our bagged DNN-based method.

Reviewers' Comments:

Reviewer #1:

Remarks to the Author:

All concerns have been addressed.